# A Review on Traversability Risk Assessments for Autonomous Ground Vehicles: Methods and Metrics

**DOI:** 10.3390/s24061909

**Published:** 2024-03-16

**Authors:** Mohamed Benrabah, Charifou Orou Mousse, Elie Randriamiarintsoa, Roland Chapuis, Romuald Aufrère

**Affiliations:** Clermont Auvergne INP, CNRS, Institut Pascal, Université Clermont Auvergne, F-63000 Clermont-Ferrand, France; charifou.orou_mousse@doctorant.uca.fr (C.O.M.); elie.randriamiarintsoa@uca.fr (E.R.); roland.chapuis@uca.fr (R.C.); romuald.aufrere@uca.fr (R.A.)

**Keywords:** risk assessment, risk metric, traversability analysis, unmanned ground vehicles, occupancy grids

## Abstract

Evaluating the risk associated with operations is an essential element of safe planning and an essential prerequisite in mobile robotics. This issue is very broad, with numerous definitions emerging in the recent literature adapting different application scenarios and leading to different algorithmic approaches. In this review, we will investigate how the state-of-the-art approaches define the traversability risk, particularly for mobile robots, whereby we classify existing risk-aware navigation algorithms according to their characterization of risk. Subsequently, we will overview the formulations of risk assessment along a path using traversability grid maps since it is essential for a mobile robot to evaluate its path to predict potential hazards. Finally, we will discuss the consistency of commonly used risk metrics in robotics. The aim of the review is to offer a comprehensive overview to newcomers in the field, to provide a structured reference for practitioners, and to also inspire future developments.

## 1. Introduction

Currently, mobile robots are deployed to accomplish different missions in unknown and partially known structured or unstructured environments (see Figure 1). These missions include search and rescue operations prompted by natural disasters [1], the inspection of planetary terrains [2], agricultural robotics [3], military surveillance missions in off-road environments [4], and autonomous driving in urban environments [5]. These different missions in such different and complex environments cannot be achieved unless a few critical issues are addressed, namely (i) perceiving the environment, (ii) characterizing and identifying potential risks and building an environment model, (iii) assessing these risks, and (iv) planning optimal safe paths while controlling the robot based on the assessed risks. Considering the complexity and diversity of the challenges associated with these sub-problems, significant research efforts have been devoted to addressing them. Focusing on the first three ones, the ability to recognize its environment and translate the data it receives from its sensors into a useful amount of knowledge is crucial for any autonomous mobile robot. This capability enables the robot to decide whether it can navigate in a safe and efficient manner or not. In reality, this problem may incorporate machine learning, feature extraction, image and signal processing, and three-dimensional (3D) geometry, thereby leading to a wide range of approaches. Given the multitude of techniques employed in terrain comprehension and the diversity of terrains in which robots operate, this review aims to summarize traversability analysis methods and their associated risk assessments.

Our paper will be organized as follows: We first set out the definitions and taxonomy of traversability analysis. Then, we provide an overview of the different approaches that roboticists use to characterize risk in their risk-aware algorithms. Subsequently, we delve into the assessment of this risk along a path in grid-based environment models (mainly occupancy grid maps). Moving forward, we discuss the various risk metrics encountered in the literature and consider that probabilistic occupancy grids treat risk as a stochastic variable. Finally, we close our study with a brief general conclusion.

## 2. Related Works and Survey Boundaries

Reviewing the field of terrain traversability analysis has been performed in numerous earlier works. In [6], the author reviewed the field of 3D terrain traversability analysis by categorizing methodologies into three main constituents, namely proprioceptive, appearance-based, and geometry-based approaches. Furthermore, ref. [7] provided a survey of the field from the perspective of planetary exploration, in which they combined pertinent applications, current methods, and underlying techniques within a unified framework. Additionally, an extensive survey of methods for planetary terrains was given in [8], where applications regarding the dry and rocky surfaces on which the planetary rovers operated were found. Advancements adopting learning-based methods to solve the problem of environment perception and interpretation with the final aim of the autonomous context-aware navigation of ground vehicles in unstructured environments were also reviewed in [9]. Ref. [10] provided a survey on sensor data fusion techniques for obstacle detection, which are applicable in off-road navigation scenarios. A more recent survey that links many of the recent advancements in traversability analysis, particularly in machine learning and semantics with classical statistical methods, was also published [11]. An overview of the aforementioned surveys is provided in Table 1.

As a complement to the aforementioned studies, the main purpose of this article is to provide a comprehensive and general overview of the traversability risk characterizations and assessments that encompass the entire range of mobile robotic applications (urban, planetary, agricultural, etc.). Our focus is to link together both classical and learning-based methods, thereby emphasizing the risk perspective rather than the methodology itself. We have also included certain points that we deemed necessary for traversability risk analysis, such as assessing the risk along a path and examining existent risk metrics.

## 3. Taxonomy

Before we tackle the question of characterizing and quantifying risk, it is imperative to outline the broad notion of risk in mobile robotics. The wide range of applications for mobile robots leads to a large variety of risks and motivates the need for a field taxonomy.

Indeed, the risks differ from one application of a mobile robot to another. For instance, the risks encountered by a rover carrying out exploration missions on Mars are not the same as those encountered by an industrial mobile robot tasked with moving pallets in a factory shed. In the first case, the robot needs to keep an eye on its power consumption and also on massive impact craters, cliffs, cracks and jagged boulders. However, the risk for an industrial mobile robot is quite different. A simple classification of these risks can be achieved by categorizing according to their source. For unmanned ground vehicles (UGVs), we can distinguish two main classes: efficiency-based and traversability-based risks. The efficiency-based category includes such elements as battery consumption [12], loss of communication [13], etc. While the traversability-based category, which is the core of our study, encompasses all the risks associated with the nature of the ground (landform, speed bumps, pot holes, slopes, etc.) and the various static or dynamic obstacles encountered by the robot in its workspace, we specifically investigated traversability-based risk in this study (Figure 2).

Up to now, traversability analysis has been widely used as a means for the optimal navigation of UGVs in environments of varying complexity to ensure safety in terms of collisions or reaching unrecoverable states. Interestingly, this generic notion of traversability has been referred to using various terms such as drivability [14], navigability [15], trafficability [16], and maneuverability [17], thereby adding complexity to its formal definition within the robotics community. For instance, in [18], robotic traversability was formalized using the psychological concept of affordances that was introduced by [19]. The authors also provided the implementation results of one part of the affordance formalism for the learning and perception of traversability affordances on a mobile robot equipped with range sensing ability, and they showed that the robot, by interacting with its environment, can learn to perceive the traversability affordances. Although this formalization seems adequate, the authors in [6] argued that it was not formalized enough to quantify traversability and to derive a continuous measure instead of a binary assessment (either traversable or non-traversable), which made it overly generic and thus incomplete. And, in turn, they defined the traversability as “the capability of a ground vehicle to reside over a terrain region under an admissible state wherein it is capable of entering given its current state, this capability being quantified by taking into account a terrain model, the robotic vehicle model, the kinematic constraints of the vehicle and a set of criteria based on which the optimality of an admissible state can be assessed” [6] (p. 2). Furthermore, the authors of [8] defined the trafficability from the terrain perspective as the terrain’s aptitude to support and provide useful traction for robot navigation. A more recent definition was proposed in [11]. In their view, by contrasting the features of the terrain to the robot’s dynamics and kinematics capabilities, traversability analyzes the terrain at a more sophisticated level than obstacle detection and this leads to the creation of an environment cost map. Many factors, including the shape of the terrain, roughness, expected friction/traction, and the vehicle’s kinematics, might be included in the study. The authors of [20] proposed an approach whereby they explicitly measured the traversal cost, as well as the associated uncertainties. They called this cost ’traversability’ (i.e., a high traversability cost corresponds to an area of the environment where the robot may suffer damage or impediments).

For us, traversability in mobile robotics refers to the ability of a robot or autonomous vehicle to navigate through a given environment effectively and safely. This notion must be understood from both perspectives: the terrain and the robot. From a terrain perspective, traversability encompasses various factors such as terrain ruggedness, obstacles, slopes, stairs, and other environmental features that may hinder or facilitate the robot’s movement. On the other hand, traversability is also dependent on the robot’s kinematics and mechanical properties, such as speed, wheel size and type, chassis height, actuators, etc.

In this paper, for simplicity, the term ’risk’ refers to traversability-based risk (mainly with respect to collision).

## 4. Traversability Risk Characterization

It is becoming increasingly crucial to assess the risk involved in a robot’s course of action as an increasing number of autonomous systems are required to carry out safety-critical missions. So, the literature includes multiple methods that are used to characterize traversability risk, and these methods have been classified in different ways in previous reviews [6,8,21]. In our case, we opted to classify them into two main categories: sensor-based and map-based. Quite simply, the difference between the two is that the first family considers risk as a constraint in a global optimization problem (typically in path planning); therefore, it exploits perceptual information (i.e., sensor data) directly without converting it into a map. Whereas, for the second category, and as its name already indicates, a traversability map of the environment (which is commonly tessellated) is created to store an estimated traversability risk that is derived from the robot’s knowledge.

### 4.1. Sensor-Based Characterization

Following the boundary of an obstacle is a standard approach used by many earlier risk-averse navigation algorithms. In most cases, the risk is simply characterized by the minimum distance to the obstacle. In [22], for example, an approach for a collision-free boundary following obstacles of arbitrary shape and globally convergent path planning was proposed using the concept of instant goals. The risk analysis was performed simultaneously and, when needed, using a vector of the measured distance of each beam of the range finder sensor. Similarly, many path planning and model predictive control algorithms generate a probational path from the obstacle representation based on the sensor data at each time step, thereby ensuring a boundary following, such as the in the Bug family algorithms [23].

In [24], a sliding mode control law was proposed that leads the robot at a pre-specified distance from the obstacle’s boundary and maintains this distance afterward. In this approach, the risk is characterized by the sliding surface, which is a function of the length of the detection ray and the tangential angle of the obstacle at the intersection point.

In another fashion, ref. [25] defined risk in a way that accounts for the size of the obstacle (agent) and the vehicle. The authors fit circles of radius *r* to an agent and limited the likelihood that their centers are inside a suitably sized “collision ellipsoid” surrounding the vehicle, which formed the basis of their risk definition. The main distinction between their approach and standard deterministic path planning formulations lies in the chance constraint. This constraint ensures that the probability of the vehicle colliding with an agent does not exceed an upper bound at each time step. The authors in [26] approximated the chance-constrained problem as a disjunctive convex program that considers polyhedral stay-in regions and polyhedral obstacles instead of circles. Furthermore, an incremental sampling-based motion planning framework was proposed in [27] through distributionally robust chance constraints, wherein they considered both sensing and localization errors, stochastic process disruptions, erratic obstacle motion, and ambiguous obstacle positions.

Since wheel–terrain interactions play a critical role in rough terrain mobility, the authors of [28] proposed a technique for accurate traversability prediction based on an online estimation of key terrain parameters using on-board sensors. In their work, the estimated terrain parameters are the cohesion and internal friction angle, which can be used to compute the terrain shear strength and thus give an estimate of robot traversability. Similarly, the authors in [29] presented a learning-based method for terrain classification into a few predefined, commonly known categories such as gravel, sand, or asphalt. Then, the traversability is measured by determining a number of parameters such as the cohesion of the soil, the internal friction angle, the radial stress, the shear displacement, etc., which is achieved using multiple sensor modalities including inertial sensors, motor sensors, range sensors, and encoders.

Exponential utility functions are also a solution to represent risk [30]. The main finding of utility theory is that there exists a utility function that converts costs into real values, or “utilities”, such that maximizing expected utility is meaningful for any attitude toward risk. In the aforementioned work, the authors chose to use the exponential utility function $u\left(c\right)=\gamma^{c}$, which was used to verify the property of “constant local risk aversion” and permitted them to express a whole spectrum of risk sensitivity. This spectrum ranges from being strongly risk-averse to being strongly risk-seeking in the function of the parameter γ. It is important to note here that *c* denotes the generalizable action cost (such as collision, resources consumption, etc.), which can be referred as the risk function for the Markov decision problem.

Since fuzzy logic has been widely applied in many fields, from control theory to artificial intelligence, it has also found its place among traversability risk analysis frameworks. In [31], the traversability risk is characterized by fuzzy rules. The authors proposed a rule-based Fuzzy Traversability Index that uses real-time measurements of the terrain attributes collected from imagery data to quantify how easy a terrain is for a mobile robot to traverse. These features include, but are not restricted to, discontinuity, slope, hardness, and roughness.

In [32], the authors represented the vehicles as particles, propagated the particles through their kinematics, and employed the resulting distribution as the risk distribution under the assumption that the geometry of the road layout is known from a high-definition map.

Quantile regression [33] is another prevalent choice to characterize risk in reinforcement learning-based frameworks. The authors in [34] introduced the Ensemble Quantile Networks (EQN) method, which combines distributional reinforcement learning with an ensemble approach to obtain a risk estimate. The distribution of risk is estimated by learning its quantile function implicitly. They demonstrated that their method can balance risk and time efficiency in different occluded intersection scenarios by considering the estimated risk. They estimated both the aleatoric uncertainty, which characterizes risk distribution (e.g., collision), and epistemic uncertainty, which arises due to the lack of knowledge and can be reduced by observing more data. In Table 2, an overview of the presented sensor-based approaches is provided, where they are further regrouped according to the targeted application and the features (criteria) used to define the traversability (Table 2).

### 4.2. Map-Based Characterization

Map-based approaches constitute another category of risk-aware navigation frameworks. As the name suggests, these techniques use an environment map as their input, which can be created from vehicle and terrain data using a variety of sensors including LiDAR, camera, IMU, GPS, and wheel odometry. Generally, the map depicts features related to terrain traversability in a tessellated fashion.

Let us start with the well-known standard of Bayesian occupancy grids [35], which model the environment by employing a probabilistic, tessellated representation of multi-source spatial data. In this framework, the map is kept as a grid of cells, each representing a small portion of the environment and possessing a value that indicates the occupancy probability of the specific element. These occupancy probabilities are updated using a Bayesian filter that treats each grid cell as an independent entity. Risk is therefore associated with the probability of occupancy. Initially developed as a 2D modeling tool, occupancy grids can be readily extended to 3D. However, 3D occupancy grids are memory-intensive, thus making them impractical for large-scale mapping applications.

In the same fashion (i.e., using the regular grid representation), many traversability maps have appeared in the literature. For instance, a traversability map was proposed by [36]. They projected the detection results of multiple sensors on a 2D probabilistic grid format, where each cell value can be understood as a probability that the vehicle can successfully drive over rather than probability occupancy (i.e., more traversability factors are taken into account such as the terrain slope). The derivation of probabilities varies depending on the sensor, with detailed steps provided for 3D LiDAR and RGB camera sensors. In [37], the authors constructed their traversability map by incorporating three fundamental terrain characteristics for each cell: the slope value, the curvature value, and the roughness. Meanwhile, in [38], the authors developed a simple traversability map where each cell of the grid contains the list of coordinates of the points falling within its bounds. Terrain classification is then performed for every cell individually using some criteria such as height variation, orientation of the vector normal to the path of terrain, and the presence of discontinuity of elevation in the cell.

In [39], the authors proposed a technique to compute a vector measure of the physical density of a given environment as perceived by each sensor modality using data from multiple sensors. Each cell of their ’density map’ consists of a feature set composed of two parts: a vector of floating point ’density’ measurements and a few auxiliary parameters. The floating point values represent the strength of the signal return from a single sensor in a single patch of the terrain. Measurements of the lowest and maximum height readings from the terrain, along with color information, are examples of auxiliary parameters.

The traversability index, firstly proposed by [40], serves to efficiently address a terrain’s ease of traversal for planetary rovers. This index represents the degree of ease with which the regional terrain could be navigated, and it is characterized by a number of fuzzy sets. In [41], the risk within each cell is expressed through its grade of membership to predefined fuzzy sets that are based on major surface features such as hills and lakes within a long range of the robot. The two main distinctions between their traversability grid and the Bayesian occupancy grid are as follows: (1) the probability of obstacle presence versus a more general concept of graded terrain quality, and (2) probability theory versus possibility theory using fuzzy sets. The authors of [42] created traversability indexes using fuzzy logic that depend on the terrain’s slope, roll variance, and roughness. They integrated fuzzy inference to combine these terrain features in order to obtain a traversability assessment and local quantitative evaluation. In [43], a method was suggested that transforms a local terrain map around the robot’s current position into a grid-type traversability map. This transformation involves extracting slope and roughness information from terrain patches using least-squares plane fitting. Subsequently, the method calculates ’polar obstacle densities’ for each cell in the traversability map and converts them into a traversability field histogram.

The 2.5D Digital Elevation Map (DEM), alternatively known as Cartesian elevation maps [44,45], is another option for traversability risk characterization as it extends the concept of 2D occupancy grids. This mapping technique stores the cell’s elevation instead of its occupancy probability. The DEM is the most widely used technique for environment modeling and traversability analysis. Therefore, several modifications have been proposed to improve its efficiency such as in [46], who considered the elevation to follow a Gaussian distribution, and the stores for each cell are not only the mean elevation, but also the variance. Although elevation maps offer compact representation, they may not adequately depict multi-level structures or even vertical structures (bridges, for instance). Therefore, the authors of [47] proposed a new representation denoted as multi-level surface maps (MLS maps), which allows one to store the elevation of multiple surfaces in each cell of the grid. However, surface representations are frequently based on significant assumptions about the corresponding environment, and they may demand a large amount of memory, especially outdoors. Additionally, they may not effectively differentiate between free and unknown space, which is crucial for safe navigation. Addressing this limitation, the authors of [48] proposed a three-dimensional model using octrees [49], thereby providing a volumetric representation of space and employing probabilistic occupancy estimation. In this model, each voxel stores a probability of occupancy that is similar to the standard Bayesian occupancy 2D grid. A visualization of the mapping results for the same environment (a tree) using some of the 3D mapping approaches mentioned in this paragraph is shown in Figure 3.

Moreover, the authors in [50] asserted that all the previously mentioned frameworks were not suitable for assessing meaningful risks, i.e., risks that keep their physical unit. Thus, they proposed a novel framework called Lambda-field, which is based on the Poisson Point Process theory. Their map stores for each cell the ‘rate’ λ of a harmful event (i.e., collision) instead of the probability of occupancy. A higher intensity λ indicates a greater likelihood of collision at a given position (see Figure 4). The Lambda-field framework permits the computation of risk along a given path while retaining its physical sense, without resorting to probabilistic forms, by simply integrating the intensity of ‘λ’. In addition, using their framework, one can choose any risk function, whether associated with the nature of the robot or that of the terrain, without any change in the theory. In their case, for instance, they take the maximum gain in kinetic energy that results from a collision as the risk function.

Furthermore, a 3D extension of the Lambda-field framework is proposed in [52], wherein they consider the time of a harmful event to be the deformation of the wheel due to a collision, and they consider the risk function as being the maximum potential energy that is absorbed by the wheels. So, they store, for each cell, the intensity of being non-crossable by the robot. In [53], we assumed that the traversability risk stems not only from collision, but also from the lack of knowledge at each position. To address this, a new map called the ’knowledge map’ was introduced, where a probability of knowledge for each cell is stored. This map was then combined with a Bayesian occupancy grid to assess the traversability risk along a path.

Learning-based cost maps have seen more recent developments for robotics and autonomous driving. In [54], an architecture was proposed wherein raw or minimally processed point cloud data are transformed and introduced into a convolutional neural network (CNN), which then generate a CVaR (Conditional Value at Risk) cost map directly. The obtained traversability cost map was more robust to outliers because of the use of CVaR, and it was more computationally efficient. In [55], the authors argued that a neural network may be trained in a self-supervised manner to examine the traversability of terrain using the empirical distribution of traction parameters in unicycle dynamics. The proposed approach takes, as the input, local semantic and elevation features to predict linear and angular traction distributions in order to generate a traction distribution map. The map indicates the reliability of the prediction, such that if the likelihood of input terrain features is below a threshold, the terrain is deemed out-of-distribution (OOD) and later avoided during planning via auxiliary penalties. In [56], the authors proposed a novel interpretation of a terrain’s traversability by learning speed and gait policies based on terrain semantics and human demonstrations. The resulting ’speed map’ provided a straightforward and intuitive understanding of the model’s predictions, and it can be used in navigational tasks such as path planning. Ref. [57] uses privileged information during training to learn navigational affordances in a modular manner, where perfect odometry availability is assumed. The authors proposed to construct a top-down belief map of the world (i.e., the mapping is driven by the needs of the planner), and they applied a differentiable neural net planner to produce the next action at each time step. In [58], the authors provided an MMP-based approach with non-linear cost functions [59], in which they integrated multiple on-board sensor features such as obstacle heights and LiDAR point densities. The learned cost map, derived from an expert’s demonstration and based on detected perceptual features, is used by the two-level planner of a six-wheeled skid steer vehicle. Table 3 summarizes the map-based traversability analysis methods discussed so far.

Despite the fact that, up to now, we have only been talking about methods that discretize the environments into cells, since this is the subject of the rest of this study, it is not the only solution available. Another solution was provided in [60], in which the environment was represented as a Gaussian process (GP). The main idea is that, for each position, the GP defines the occupancy as a Gaussian distribution, one that is characterized by a mean μ and an associated predictive variance σ. This utilization of a Gaussian process entails a non-parametric Bayesian learning technique, thereby enabling the exploitation of correlations between points on the map. This aspect is not addressed by mapping techniques like the occupancy grids mentioned earlier. To answer the drawbacks of the GP approach, the authors of [61] proposed a simpler and faster approach for environment representation through continuous occupancy mapping. The technique, named Hilbert maps, represent the occupancy property of the world with a linear discriminative model operating on a high-dimensional feature vector that projects observations into a reproducing kernel Hilbert space. Just like the GP, this approach provides maps at any resolution since it does not presuppose that the world is divided into grid cells and naturally captures the spatial correlations between measurements.

Until now, our discussion has revolved around risk mapping methods tailored for static environments. However, traversability risk may also arise from dynamic obstacles, as previously mentioned. In this context, the issue of creating continuous occupancy maps of dynamic environments for robotics applications is addressed in [62] by learning a kernel classifier on an efficient feature space. They incorporated the temporal variations into the spatial domain by propagating motion uncertainty into the kernel using a hierarchical model that enables a direct prediction of future occupancy maps from past observations. Although fast optimization methods and heuristic tuning were required, ref. [63] proposed a Bayesian approach to address this issue, thus eliminating the regularization term of Hilbert maps. They extended this approach to deal with dynamic environments by leveraging neighborhood information to reduce susceptibility to occlusions. While the initial formalization of Hilbert maps may not fully support their methods, their implementation underscores the utility of Bayesian methods for dynamic environments. In a more recent work [64], the authors proposed a framework that allows for a real-time computation of the previous approach. It aims to capture the uncertainty of moving objects, and this uncertainty information is then incorporated into the Hilbert maps. Nevertheless, their models currently struggle to track complex patterns and only deal with simple motion models. Additionally, ref. [65] addressed the dynamic occupancy grid issues by presenting a stereo-vision-based framework. It consists of two components: a motion estimation for both the moving ego-vehicle and the independent moving objects; and dynamic occupancy grid mapping, which relies on the estimated motion information and dense disparity maps. Despite its ability to map occupied areas and moving objects concurrently, further optimization is needed, especially when multiple objects are present, to meet real-time constraints.

Another group of frameworks [66,67,68] offered an alternative approach for dealing with dynamic and occluded objects. In [66], Markov chains were used to model the dynamics of moving pedestrians and to predict their potential future locations. These occlusion estimations are mapped into risk regions and then employed to plan a path through a potentially obstructed area. Similarly, ref. [68] introduced the same idea. By analyzing the scene and evaluating the best possible future behavior, the authors proposed to map the expected risk according to the dynamic variations of the acting ’entities’. Their predictive risk map indicates how risky a certain behavior will be in the future and enables the evaluation of risk-minimizing behavior on path planning.

With the rise of neural networks, some of the literature [69,70] advocates for machine learning approaches in dynamic traversability mapping. The authors of [69] tackled the velocity estimation errors highlighted in [71] using a recurrent neural network architecture. Meanwhile, ref. [70] tackled the issues related to dynamic occupancy grid maps by combining a Bayesian filtering technique and a deep convolutional neural network. These works provided efficient segmentation of the static and dynamic zones. However, the performance of such methods sorely depend on the dataset used for training. More generally, the big drawback of learning methods is the training step.

## 5. Risk Assessments along a Path in Traversability Grid Maps

In the preceding section, we provide a comprehensive overview of traversability risk characterizations in terrestrial mobile robot navigation frameworks. These are classified into two main categories: sensor-based and map-based. In this section, our focus shifts to the second category, which more specifically assesses the risk along a path using traversability grids. These metric maps are particularly well suited for risk assessment since they tessellate the environment into cells where each one stores the information about potential risky event (such as collision, elevation, roughness, lack of knowledge, etc.). Given that the environment, represented as a field $e:R^{n}\to R$, possesses an infinite number of degrees of freedom, which makes it non-storable on a machine, tessellating the environment into cells of fixed size and assuming that the field is constant inside the cells is the most appropriate solution for assigning an occupation to each position in the environment.

In [72], the authors used the occupancy grid to simply make a binary classification of the cells (either occupied or free according to a predefined threshold). Subsequently, each cell was assigned a reward according to its occupancy status, with positive values denoting occupied cells and negative values indicating free cells. By summing up the rewards of all the cells along a given path, they could obtain an occupancy cost for the path. This allows for a comparison of the various trajectories from a list of tentacles, where the safest trajectory is selected as the one yielding the highest reward.

If we continue to utilize the standard Bayesian occupancy grids, and if we reduce for now the risk to the probability of collision, the risk of crossing a path $P_{\left[0,i\right]}$ given by a set of *i* cells is as follows:(1)$$R\left(P_{\left[0,i\right]}\right)=1-\prod_{j=0}^{i}\left(1-P_{j}\right),$$
where $P_{j}$ is the probability of occupation of the $j^{th}$ cell.

Another probabilistic definition of risk is ’the probability of the robot not being able to finish the path’, which was proposed in [73]. Its expression is given by the following:(2)$$R\left(P_{\left[0,i\right]}\right)=1-\prod_{j=0}^{i}\prod_{k=0}^{r}\left(1-r_{k}\right),$$
where $r_{k}$ is the probability of occupied cell *k* causing failures at state *j* given the history of finishing the cells with indexes from 0 to *j*.

In [74], the authors assumed that Equation (Equation 1) is valid only if $P_{j}$ is defined as the probability of a collision to occur within the corresponding time interval and that it can possibly normalized as such; otherwise, the definition of $R(P_{\left[0,i\right]})$ would depend on the time step. Thus, they defined a temporal risk function called TTC (Time To Collision) $\tau (P_{\left[0,n\right]})$ as the expectancy of the time of the first collision. Its expression is given by Equation (Equation 3).
(3)$$\tau \left(P_{\left[0,n\right]}\right)=t_{0}R\left(P_{0}\right)+\sum_{i=1}^{n}t_{i}R\left(P_{\left[0,i\right]}\right)\left(1-R\left(P_{\left[0,i-1\right]}\right)\right)$$

Although Equation (Equation 1) seems very reasonable, because the risk here is none other than the probability of colliding at least one cell in the path (which is the complement of the joint probability that all cells are free), the authors in [50] were able to show that the above calculation is ill-formed. This was illustrated using the example depicted in Figure 5.

In this case, if we simply apply the above equation to evaluate the risk over the two (orange and blue) paths leading to the objective, we will find that the probability of collision of the orange one has a value of 0.97, while that of the blue is 0.99. As such, instead of choosing the blue path, which seems intuitively more safe, the robot would take the orange one, which passes through a large number of highly occupied cells. Additionally, they highlighted that the probability of collision was significantly influenced by the discretization. An illustrative example is shown in Figure 6.

Through this example, we can see that the two scenarios produced different collision probabilities, whereas the robots crossed the same uniform probability field.

These limitations prompted the authors to find more appropriate solutions. For example, the authors in [75] considered the risk over a path to be a measure of the expected loss. The expectation, in this case, was taken over the probability that the vehicle will collide with an obstacle at cell *c*. Hence, the risk is expressed by the following:(4)$$R\left(P\right)=\sum_{i=0}^{N-1}P_{i}\cdot L\left(x_{v},x_{c}\right),$$
where $L(x_{v},x_{c})$ is the loss as a function of the vehicle state $x_{v}$ and the cell state $x_{c}$. In their work, they defined risk as the total loss of kinetic energy of the system. This was also the same for [53], wherein they defined risk as the integral over the trajectory (expected value) of the kinetic energy of contact, which is weighted by a probability of risk.

Indeed, in this case, risk *R* is not a probability of collision as it lies in [0,∞), but rather it represents a kinetic energy expressed in (*Joule*), i.e., this time it has a physical meaning but is still dependent on the tessellation size—the smaller the size of the cells, the more cells the robot lies on and therefore the larger *R* is.

Furthermore, we can consider the case represented by Figure 7, where the robot needs to choose between the two paths (top and bottom) by assessing the risk using Equation (Equation 4).

If we consider that the robot has a mass *m* and runs at constant speed *v*, it becomes evident that both trajectories would have the same risk value. However, it is apparent that the second trajectory was more risky due to an 80% occupied cell.

The risk was therefore multiplied by the cell area Δa, as one suggested a solution to the cell size dependency issue.
(5)$$R\left(P\right)=\Delta a\sum_{i=0}^{N-1}P_{i}\cdot L\left(x_{v},x_{c}\right)$$

But, by multiplying by the area, we lose the physical meaning of R(P) (i.e., this is leading to the risk being expressed in $J\cdot m^{2}$).

Another solution to infer the probability of collision in occupancy grids was proposed in [76]. They proposed a new interpretation of risk that accounts for the fraction of time that a robot has stayed in a cell while following a trajectory using the theory of product integrals [77]. The probability of collision based on this theory is then given by the following:(6)$$R\left(P\right)=\prod_{0}^{L}\left[1-p_{0}\left(P\right)\right){]}^{ds},$$
where $p_{0}$ denotes the probability of collision over P. We should note here that one of the drawbacks of their approach is that the robot is reduced to a point (i.e., the robot’s wheelbase is zero).

Therefore, the probability of collision does not rely on the size of the tessellation when we integrate over the tessellated field.

In addition, the authors in the already-cited paper [20] used the time consistency dynamic risk measure [78], which is defined for a set of states $x_{0:N}$ and a policy π, as per the following:(7)$$J\left(x_{0},\pi ;m\right)=R_{0}+\rho_{0}\left(R_{1}+\rho_{1}\left(R_{2}+...+\rho_{N-1}\left(R_{N}\right)\right)\right),$$
where ρ denotes the conditional value at risk $\left(CVaR_{\alpha }\right)$, which was discussed in the following section; and $R_{i}$, which depicts the risk for time *i*. Moreover, for α→0, the risk function *J* simplifies to
(8)$$\begin{array}{cc} \lim_{\alpha \to 0}J\left(x_{0},\pi ;m\right) & =R_{0}+E\left[R_{1}+E\left[R_{2}+\ldots +E\left[R_{N}\right]\right]\right],\\ \; & =R_{0}+\sum_{i=1}^{N}E\left[R_{i}\right]. \end{array}$$

One can easily notice that the smaller the discretization, the higher the risk (i.e., when the number of sample times increases to infinity, the risk tends to infinity).

When discussing the dependence on cell size and the physical sense of risk, we cannot avoid citing [50]. In that study, the authors proposed a new framework for not only representing cell occupancy differently (as mentioned in the previous paragraph), but also for assessing physical risk over a path. They defined the probability of encountering at least one collision on the path as follows:(9)$$R\left(P\right)=\int_{P}f\left(a\right)da≃1-\exp(-A\sum_{c_{i}}\lambda_{i}),$$
where f(a) is the probability distribution associated to the Lambda-field framework as a function of the traveled area *a*. Each cell $c_{i}$ has an area *A* and an associated lambda $\lambda_{i}$.

In addition, by choosing a physical risk function (force of collision, energy, etc.) r(·), they defined its expectation over the path P going through the cells ${\left\{c_{i}\right\}}_{0:N}$ by the following:(10)$$E\left[r\left(X\right)\right]=\sum_{i=0}^{N}r\left(Ai\right)\exp\left(-A\sum_{j=0}^{i-1}\lambda_{j}\right)(1-\exp(-A\lambda_{i}),$$
where *X* is the position (i.e., area) at which the first event ‘collision’ occurs. Although the Lambda-field framework is not yet popular among the robotics community, it remains a very suitable choice for assessing the physical risk along a path.

## 6. Quantification Metrics for Stochastic Risk

In most cases, and especially in grid-based maps, the risk is a stochastic variable. A map is fundamentally, most of the time, a field of binary random variables. A sensor is a probabilistic channel that links robot motion in the physical world to information gain. Quantifying risk, then, corresponds to evaluating a risk metric, i.e., a mapping from the cost random variable to a real number.

As such, ref. [79] provided an exhaustive study of what constitutes a ’good’ risk metric by advocating axioms that must be fulfilled by the latter:A1. Monotonicity;A2. Translation invariance;A3. Positive homogenity;A4. Subadditivity;A5. Comonotonic additivity;A6. Law invariance.

The expected value used, for instance, in [50] describes the long-term average level of risk based on its probability distribution. It constitutes a simple and coherent metric in the sense that it satisfies all of the axioms of [79], as well as the worst case metric [80], which refers simply to the largest value of risk. Another popular metric to quantify risk in robotics applications is the mean variance E[X]+βVar[X], which is used—for instance—in [81]. The mean variance satisfies only the axiom ’A6’, thereby rendering it a non-coherent metric.

In addition to statistics, finance specialists also have their own risk metrics, which are increasingly used in robotics. One should start with the most popular Value at Risk (VaR), which is defined as the (1−α)-quantile of the cost distribution. Its expression is given in Equation (Equation 11).
(11)$${\mathrm{VaR}}_{\alpha }\left(X\right)=min\left\{x\;|\;P\left(X>x\right)\leq \alpha \right\}.$$

In actuality, VaR is a non-coherent risk metric since it satisfy only five axioms. In line with the aforementioned chance constraints [25,26,27], the VaR was found to be closely related since the constraint $Var_{\alpha }\left(Z\right)\leq 0$ corresponded to the chance constraint P(X>0)≤α.

The expected shortfall, or the Conditional Value at Risk (CVaR) metric [82], was introduced in several risk-aware path planning applications [20,83] as an alternative to VaR. Intuitively, CVaR is the expected value of costs in the conditional distribution of the cost distribution’s upper (1−α)-tail; thus, it is a metric of ’how bad is bad’. Moreover, the CVaR can be expressed as a function of VaR as follows:(12)$${\mathrm{CVaR}}_{\alpha }\left(Z\right):=\frac{1}{\alpha }\int_{\left(1-\alpha \right)}^{1}VaR_{1-\tau }\left(Z\right)\;d\tau .$$

As we said before, VaR has some shortcomings and that is why CVaR, which is a coherent risk metric, comes in handy.

Entropic Value at Risk (EVaR) [84] is another financial risk metric that has been adopted for mobile robot risk assessment applications [85]. EVaR is an upper bound for both the value at risk (VaR) and the conditional value at risk (CVaR), which is obtained from the Chernoff inequality.
(13)$$EVaR_{1-\alpha }\left(X\right)=\underset{z>0}{\inf}\left[z^{-1}\ln\frac{E[e^{Xz}]}{1-\alpha }\right].$$

Even though both the EVaR and the CVaR are coherent metrics, EVaR is a more risk-sensitive measure.

A comparison of the most used risk metrics is illustrated in Figure 8.

In [87], the authors presented a new risk measure that is a generalization of EVaR called Relativistic Value at Risk (RLVaR). It is a special case of ϕ−divergence risk measures based on Kaniadakis entropy. The RLVaR is a coherent risk metric that has not yet been adopted in robotics.

An overview of the discussed risk metrics is provided in Table 4, which was achieved by determining their coherence according to [79].

## 7. Conclusions and Future Works

In this review, we provided a glance at the main methods that are to define and assess risk in mobile robotics applications. The risk characterization approaches have been divided into two main categories: sensor-based and map-based methods. In contrast to the second category, which does require transforming sensor data into an environment map (mainly in grid format), the first family includes all of the methods that utilize sensor data without such transformations. Map-based methods produce accurate metric maps that encode the traversability information and allow for efficient risk-aware planning. However, this efficiency decreases in large-scale indoor environments. Sensor-based methods also enable efficient reactive navigation but lack a suitable model of the environment. We then focused our study on the second category by dealing with the assessment of traversability risk along a path in such traversability grids. Several formulations were discussed, wherein we highlighted their drawbacks and used some examples. We opted to end this mini-review by discussing the various existing risk metrics, which represent an important field in robotic risk assessment applications. Thus, investigating these risk metrics led to the conclusion that not all of them exhibit coherence, and certain metrics stand out as objectively superior to others.

The present paper has outlined several major advances and breakthroughs in the field of traversability analysis. This represents a solid foundation for our future work, where the aim is to develop a risk-aware navigation framework that enables an autonomous vehicle to operate efficiently in urban environments with numerous sources of information available, as well as in off-road or rural zones. To achieve this objective, a traversability analysis must be carried out that takes into account all of the aspects ranging from the nature of the terrain, through the robot’s mechanical limitations, to an analysis of the robot’s knowledge (i.e., perception capabilities and information availability), as initiated in our earlier work [53].

## Figures and Tables

**Figure 1 sensors-24-01909-f001:**
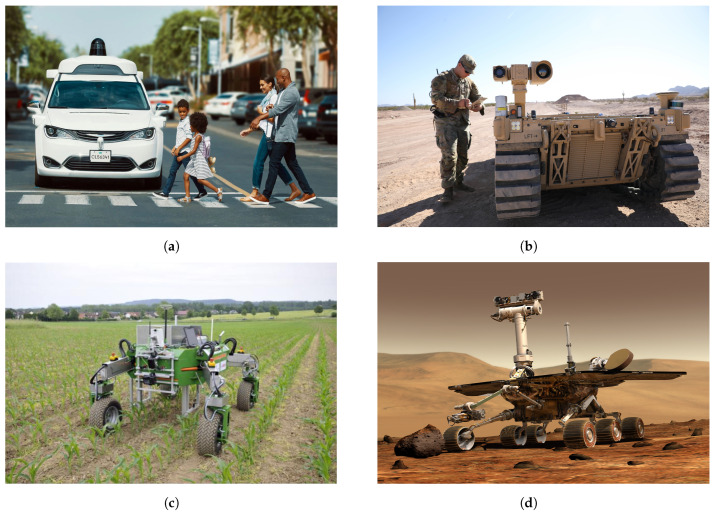
Autonomous ground vehicles applications. (**a**) self-driving cars; (**b**) military autonomous robot; (**c**) agricultural autonomous robot; and (**d**) the Mars rover.

**Figure 2 sensors-24-01909-f002:**
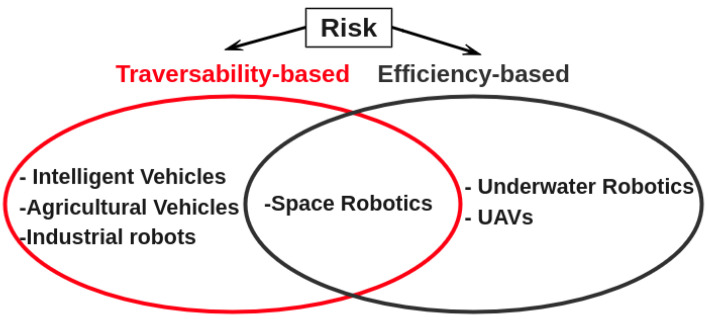
Risk types with examples of robotic applications.

**Figure 3 sensors-24-01909-f003:**
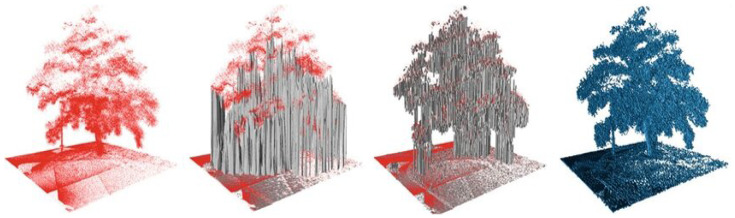
Three-dimensional representations of a tree scanned with a 3D LiDAR (from **left** to **right**): point cloud, elevation map, multi-level surface map, and octomap. From [48].

**Figure 4 sensors-24-01909-f004:**
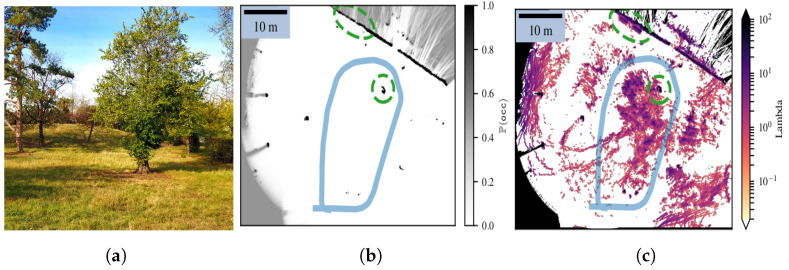
Mapping of an unstructured grassed zone (**a**) using both a Bayesian occupancy grid (**b**) and the Lambda-field framework. (**c**) The robot, with its path in light blue, went around the nearest tree (circled in green) before going back to its initial position (the path in blue). From [51].

**Figure 5 sensors-24-01909-f005:**
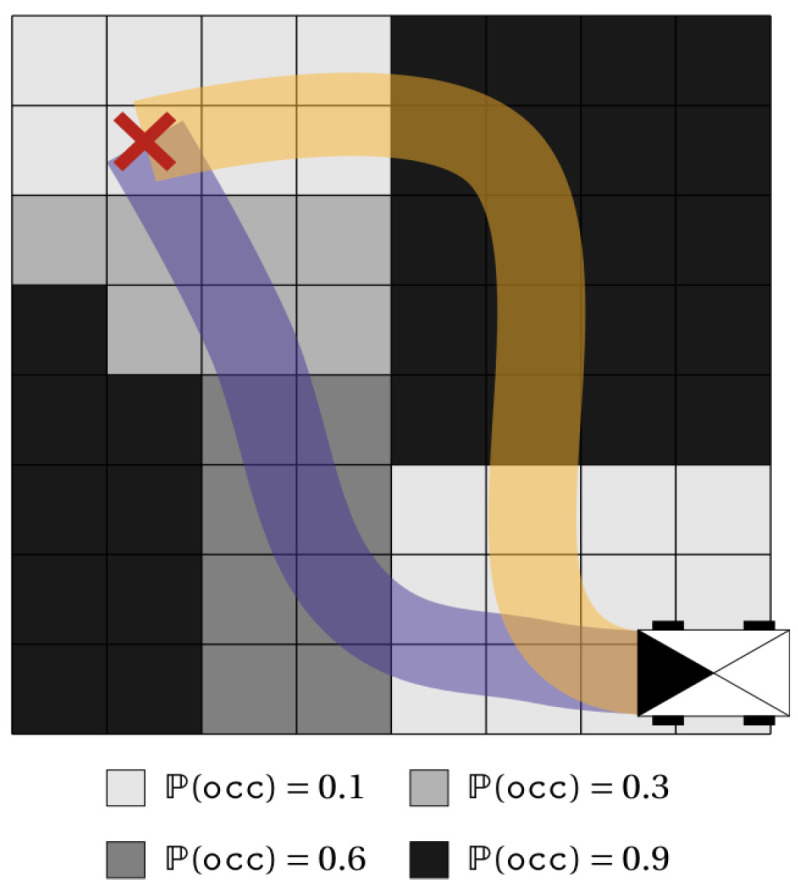
Example of an occupancy grid where the robot needs to cross and reach the goal in red, with two possible paths in blue and orange. From [50].

**Figure 6 sensors-24-01909-f006:**
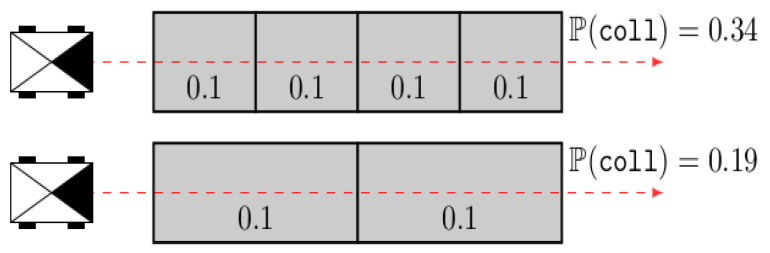
The robots, which are represented as boxes with a filled triangle on the front, aim to navigate through an environment by following the dashed red line. The collision probability is uniform for the whole environment.

**Figure 7 sensors-24-01909-f007:**
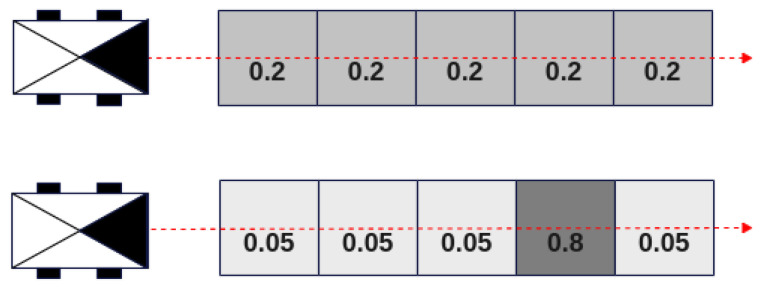
Example of risk assessments using Equation (Equation 4).

**Figure 8 sensors-24-01909-f008:**
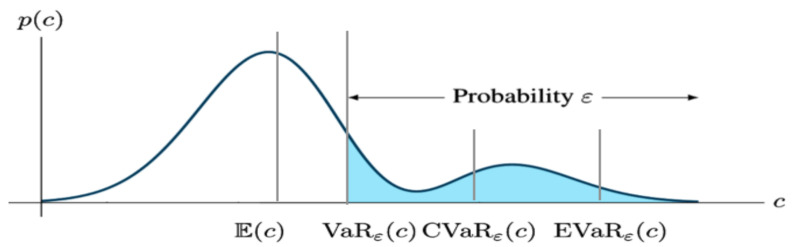
Comparison of the mean, VaR, CVaR, and EVaR (obtained from [86]). The axes denote the values of risk *c* and its probability density function p(c). The blue area denotes the (1−α)% of the area under p(c).

**Table 1 sensors-24-01909-t001:** Overview of the related surveys.

Paper	Year	Description	Contribution
Sancho-Pradel and Gao [7]	2010	Planetary exploration	A survey of the field from a planetary exploration perspective, bringing together the underlying techniques, existing approaches, and relevant applications under a common framework
Chhaniyara et al. [8]	2012	Planetary exploration	Brought together vital information pertaining to various terrain characterization techniques into a single article
Papadakis [6]	2013	Universal	Reviewed the field of 3D terrain traversability analysis by aggregating the diverse contributions from individual domains and elaborating on a number of key similarities, as well as differences
Guastella and Muscato [9]	2020	Unstructured Environments	Reviewed the contributions that adopted learning-based methods to solve the problem of environment perception and interpretation with the final aim of the autonomous context-aware navigation of ground vehicles in unstructured environments
Hu et al. [10]	2020	Obstacle detection	Summarized the considerations of the onboard multi-sensor configuration of intelligent ground vehicles in off-road environments
Borges et al. [11]	2022	Universal	Reviewed the literature of terrain traversability analysis and defined unambiguous key terms, as well as discussed the links between the fundamental building blocks that range from terrain classification to traversability regression

**Table 2 sensors-24-01909-t002:** Overview of sensor-based approaches.

References	Method	Traversability Risk	Application	Criteria
[22]	Instant goal	Minimum distance to obstacle	Obstacle avoidance	Obstacle
[23]	μNav	Minimum distance to obstacle	Obstacle avoidance	Obstacle
[24]	Sliding surface	Breach a set distance to obstacle	Obstacle avoidance	Obstacle
[25,26,27]	Chance constraint	Probability of collision	Obstacle avoidance	Obstacle/Robot
[28,29]	Proprioceptive sensing	Terrain parameters	Off-road navigation	Terrain/Robot
[30]	Exponential utility functions	Unspecified cost function	Universal	Unspecified
[32]	Particle filtering	Particle distribution	Navigation under occlusions	Obstacle
[31]	Fuzzy rules	Membership to Fuzzy Traversability Index	Off-road navigation	Terrain
[34]	Quantile regression	Uncertainties	Navigation under occlusions	Obstacle/Robot

**Table 3 sensors-24-01909-t003:** Overview of map-based approaches.

References	Risk Characterization	Map Dimensions	Paper Application
[35]	Probability of occupancy	2D	Universal
[36]	Probability of traversability	2D	Off-road navigation
[37]	Slope, curvature, and roughness	2.5D	Off-road navigation
[38]	Binary classification	2D	Off-road navigation
[39]	Object density	2D	Off-road navigation
[41]	Degree of membership to fuzzy sets	2D	Off-road navigation
[44,45]	Elevation	2.5D	Universal
[47]	Elevation	2.5D	Environments with vertical structures
[48,49]	Probability of occupancy	3D	Universal
[50,52]	Rate of a harmful event	2D	Off-road navigation
[54]	CVaR of unspecified variable	2D	Off-road navigation
[55]	Traction distribution	2D	Off-road navigation
[56]	Speed	2D	Off-road navigation
[58]	Generalizable traversability cost	2D	Complex unstructured terrain
[60]	Gaussian distribution	2D/3D	Urban environment
[61]	Probability of occupancy	2D/3D	Urban environment

**Table 4 sensors-24-01909-t004:** Coherence of the mentioned risk metrics.

Metric	Axioms [79]	Coherence
Expected Value	A1–A6	Coherent
Worst Case	A1–A6	Coherent
Mean Variance	A6	Non-coherent
Value at Risk (VaR)	A1–A3, A5, A6	Non-coherent
Conditional Value at Risk (CVaR)	A1–A6	Coherent
Entropic Value at Risk (EVaR)	A1–A6	Coherent
Relativistic Value at Risk (RLVaR)	A1–A6	Coherent

## Data Availability

The data are contained within the article.

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
