# Peer review of "A Review on Traversability Risk Assessments for Autonomous Ground Vehicles: Methods and Metrics"

_sensors, 2024, doi:10.3390/s24061909_

Round 1

Reviewer 1 Report

Comments and Suggestions for Authors

This review paper provides a good summary of the process for evaluating the terrain risk for mobile, autonomous robots. The writers' extensive experience is evident in the way they have reviewed, categorized, and analyzed pertinent research and methodologies, thoroughly reflecting the state of the field's research. For academics conducting related research, it is an excellent resource. In my capacity as a reviewer, I suggest that the journal publish this manuscript.

Comments on the Quality of English Language

Please carefully review phrases in English to ensure that spelling and grammar are correct.

Author Response

Dear Reviewer,
Thank you for giving us the opportunity to submit a revised draft of my manuscript titled “A review on traversability risk assessment for autonomous ground vehicles: Methods and metrics” to the special session “Intelligent Transportation Systems: Sensing, Automation and Control” of physical sensors journal. We appreciate the time and effort that you and the reviewers have dedicated to providing your valuable feedback on our manuscript. We are grateful to you for your insightful comments on our paper. We have been able to incorporate changes to reflect your suggestions about grammar and spelling mentioned in your comment:
“Please carefully review phrases in English to ensure that spelling and grammar are correct”
Please be assured that each of us -the authors- has done his best revising the grammar and the spelling of every sentence of the article. In addition, all spelling and grammatical errors pointed out by the reviewers have been corrected.
We look forward to hearing from you in due time regarding our submission and to respond to any further questions and comments you may have.

Yours sincerely,

Mohamed BEN RABAH
March 7 th , 2024

Reviewer 2 Report

Comments and Suggestions for Authors

In this review, the authors investigated how the state of the art defines the traversability risk, for mobile robots in particular, and thereby classify existing risk-aware navigation algorithms according to how they characterize risk. They also made an overview about the formulations of risk assessment along a path using traversability grid maps since it’s essential for a mobile robot to evaluate its path to predict potential hazards. Finally, the authors discussed the consistency of the commonly used risk metrics in robotics.

Paper is in readable form and the authors are advised to address the below concerns and I recommend minor revision.

1. Survey boundaries to be summarized in a table for easy understanding to the readers as this is a survey paper.

2. Very few references are taken between 2020 to 2023 which may leads to missing state of the art works.

3. Authors considered only traversability and not the efficiency based as they want to have depth analysis on traversability. But still, few more factors with respect to traversability is to be considered in table 1.

4. Why no experimentation is done and not reported about the traversability risks?

5. Section 5 is good.

6. Future work to be indicated in last section.

Comments on the Quality of English Language

Minor editing of English language required

Author Response

Dear Reviewer,
Thank you for giving us the opportunity to submit a revised draft of my manuscript titled: “A review on traversability risk assessment for autonomous ground vehicles: Methods and metrics” to the special session “Intelligent Transportation Systems: Sensing, Automation and Control” of physical sensors journal. We appreciate the time and effort that you and other reviewers have dedicated to providing your valuable feedback on our manuscript. We are grateful to you for your insightful comments on our paper. We have been able to
incorporate changes to reflect most of your suggestions.
Here is a point-by-point response to your comments and concerns:
• Comment 1: Survey boundaries to be summarized in a table for easy understanding to the readers as this is a survey paper.

Response: Indeed, you are right that a table gives a better overview of the survey boundaries, so we have added a table summarizing the application and the contribution of each paper. Please refer to table 1 in page 3/19.

• Comment 2: Very few references are taken between 2020 to 2023 which may leads to missing state of the art works.
Response: This comment is quite interesting, as it's true that recent contributions should be put forward, but we can assure you that 24 out of 87 references are between 2020 and 2023, which makes a ratio of around 27%. So, we don't think there will be any missing state of the art.

• Comment 3: Authors considered only traversability and not the efficiency based as they want to have depth analysis on traversability. But still, few more factors with respect to traversability is to be considered in table 1.
Response: Responding to this comment, we have added a column to the table to further specify the features considered to define traversability (Robot constraints, terrain properties or obstacles). Please refer to table 2, page 6/19.

• Comment 4: Why no experimentation is done and not reported about the traversability risks?
Response: For this point, we would like to clarify that the the article is a review, and its aim is to summarize existing and relevant approaches of traversability risk analysis, not to compare them. In addition, it's hard to find a criterion or metric to evaluate or compare the approaches, but it would have been easier if we had included the path planning aspect in our study. In addition, it is certain that in future work there will be experimental results and perhaps comparisons between the selected approaches for our framework.

• Comment 5: Section 5 is good
Response: No response expected by this comment, but we would like to thank you for the positive feedback.

• Comment 6: Future work to be indicated in last section.
Response: Thank you for reminding us of this important point, and we would like to inform you that we have added a paragraph to the conclusion, providing a brief description of the future work carried out by the team. Please refer to page 15/19 from line 523.

In addition, all spelling and grammatical errors pointed out by the reviewers have been corrected.

Yours sincerely,
Mohamed BEN RABAH

March 7 th , 2024

Reviewer 3 Report

Comments and Suggestions for Authors

The submitted article presents a review on traversability risk assessment on GVs, stating the current used methods and metrics. 

The article has 44% of match with other documents in the literature, so this must be changed to stay under the recommended 30%. 

It is a well described review, not a systematic, thus it don't need to have metrics describing the searches. 

In my opinion, this article will be interesting for those who start working with these technologies. 

Comments on the Quality of English Language

English language not worth mentioning.

Author Response

Dear Reviewer,
Thank you for giving us the opportunity to submit a revised draft of my manuscript titled “A review on traversability risk assessment for autonomous ground vehicles: Methods and metrics” to the special session “Intelligent Transportation Systems: Sensing, Automation and Control” of physical sensors journal. We appreciate the time and effort that you and other reviewers have dedicated to providing your valuable feedback on our manuscript. We are grateful to you for your insightful comments on our paper.
Responding to your comment:
“The article has 44% of match with other documents in the literature, so this must be changed to stay under the recommended 30%”
Please make sure that all paragraphs that match with other papers have been reformulated. Except for the paragraph between quotation marks in page 4/19 (line 93 to 97), which is a definition originally provided by the authors that we have decided to maintain refer to the page of the article from which it is taken.

For the rest of the comments, we think that no further response is expected, but we would like anyway to thank you for the positive feedback.

In addition, all spelling and grammatical errors pointed out by the reviewers have been corrected.

Yours sincerely,

Mohamed BEN RABAH
March 7th, 2024

Reviewer 4 Report

Comments and Suggestions for Authors

The article presents a review on traversability risk assessment for autonomous ground vehicles. The review is vastly documented and analysis is properly conducted.

Some minor suggestions:

Please check the spelling within the document (eg. Line 108 “will will”)

Even though each method is thoroughly described maybe the authors can provide the number of articles used, inclusion/exclusion criteria, libraries used.

Maybe the authors can state the objective of their study, haw will this study be used?

Best regards

Comments on the Quality of English Language

there are some repetitions in the text : line 108 "will will"

Author Response

Dear Reviewer,
Thank you for giving us the opportunity to submit a revised draft of my manuscript titled “A review on traversability risk assessment for autonomous ground vehicles: Methods and metrics” to the special session “Intelligent Transportation Systems: Sensing, Automation and Control” of physical sensors journal. We appreciate the time and effort that you and other reviewers have dedicated to providing your valuable feedback on our manuscript. We are grateful to you for your insightful comments on our paper. We have been able to incorporate changes to reflect most of your suggestions.
Here is a point-by-point response to your comments and concerns:

• Comment 1: Please check the spelling within the document (eg. Line 108 “will will”).
Response: You are right that there were quite a few spelling and grammar mistakes, and we thank you for pointing it out. Please be assured that each of us -the authors- has done his best revising the grammar and the spelling of every sentence of the article. In addition, all spelling and grammatical errors pointed out by the reviewers have been corrected.

• Comment 2: Even though each method is thoroughly described maybe the authors can provide the number of articles used, inclusion/exclusion criteria, libraries used.
Response: The bibliographical research carried out as part of this study included more than 120 articles which were filtered to meet the purpose of the study, which aimed to deal with the analysis of traversability from a risk perspective, and also to fit in with the categorization we have made. Furthermore, since the article does not present any experimental results, no specific libraries or datasets were involved.

• Comment 3: Maybe the authors can state the objective of their study, haw will this study be used?
Response: Thank you for reminding us of this important point. Indeed, we have added some elements to the abstract to highlight the utility and the purpose of the study. Please refer to the abstract in page 1/19 (line 9 to 10).

Yours sincerely,

Mohamed BEN RABAH
March 7th, 2024

Round 2

Reviewer 3 Report

Comments and Suggestions for Authors

Thank you for considering the suggestions and changed the document.